# The Strength of Pine (*Pinus sylvestris* L.) Sawn Timber in Correlation with Selected Wood Defects

**DOI:** 10.3390/ma15113974

**Published:** 2022-06-02

**Authors:** Marek Wieruszewski, Adrian Trociński, Jakub Kawalerczyk, Adam Derkowski, Radosław Mirski

**Affiliations:** Department of Mechanical Wood Technology, Poznan University of Life Sciences, Wojska Polskiego 28, 60-627 Poznan, Poland; adrian.trocinski@up.poznan.pl (A.T.); jakub.kawalerczyk@up.poznan.pl (J.K.); adam.derkowski@up.poznan.pl (A.D.); radoslaw.mirski@up.poznan.pl (R.M.)

**Keywords:** structural timber, pine sawn timber, wood defects, modulus of elasticity, strength properties

## Abstract

Pine timber of Polish origin intended for structural purposes is characterized by significant variability in the quality parameters. Technological suitability determined on the basis of relevant international classifications is based on the assessment of both selected mechanical and physical properties of wood. Moreover, the description of visual properties is also a valuable indicator regarding defect distribution. In the group of quality features playing a crucial role in the classification of sawn timber, there are knots, disruptions of grains, cracks, etc. Thus, the aim of the research was to determine the correlation between the presence of selected defects and the strength properties of individual timber pieces. This type of study is based on a nondestructive test method that allows for high optimization of sawn materials processing. In the case of sawn timber of Polish origin, the modulus of elasticity (*MOE*) determined using the sonic test is commonly used as a criterion. The research material was harvested from southern Poland. The results of the conducted studies confirmed a correlation between an increasing occurrence of particular types of defects and the results of *MOE*. Furthermore, as a result of the performed investigations, no significant effect of narrow surface cracks on strength properties was observed.

## 1. Introduction

The suitability of sawn materials is strongly dependent on the structure of the wood and the frequency of wood defect occurrence. The features commonly used to characterize both the cross-section and the external shape of the log can often translate into the strength characteristic affecting its suitability for structural application. The proper management of cutting processes indicates which quality standards should be applied depending on the wood species, its dimensions, and its origin. Proposing the appropriate optimization model is crucial for wood sorting and helps to solve the problem of planning the production of construction sawn timber [1,2]. The currently applied optimization systems for sorting structural timber (e.g., MiCROTEC and LuxScan) are based on the identification of defects and their acceptability in various strength functions created for semifinished products. Only models that focus on maximizing the use of information concerning the quality characteristics of a given wood species guarantee the achievement of the finished product with an appropriate strength class and overall assumed production results. 

The timber assessment systems used to ensure optimal wood processing into structural semifinished products are most often based on visual assessment, determination of the dynamic modulus of elasticity, or both methods combined together. The identified anatomical defects, which include, among others, knots, slopes of grain, and resinosis, play a decisive role in classification systems as well as defects occurring due to secondary processing, i.e., cracks, discolorations, rots, and curvatures [3,4]. In the group of the abovementioned coniferous wood defects, knots most often play a crucial role in the process of optimizing the obtainment of structural semifinished products. The results have shown that in the case of coniferous wood species, knots constitute up to 75% of defects, affecting the qualitative classification of the obtained sawn timber [5]. At the same time, the presence of knots strongly influences the strength of the wood determined using the modulus of elasticity [6]. This parameter is especially important since it is the basis for the comparative assessment of the technical value of wood originating from Poland [7,8,9,10,11]. Knots disturb the homogeneity of the wood structure by a local increase in hardness and density and a change in the direction of the fibers [12,13,14]. Moreover, the presence of knots negatively affects tensile, longitudinal compression, and static bending strength when compared with knot-free wood. The deterioration in strength strongly depends on the dimension, soundness, and distribution of knots within the investigated element.

The horizontal cracks occurring along the length of the sawn timber piece also play a significant role in terms of strength properties [15,16]. In the case of surface assessment, an important role is played by the area of the cracks characterized based on their width and length. However, the verification of depth is the most important for predicting a decrease in strength properties. In general, as the depth and length of the cracks increase, deterioration in the loads transferred by structural materials also increase [17].

The slope of the grains in wood materials indicates the deviation of the anatomical elements of wood from the longitudinal axis of the sawn timber [18]. This defect in the wood structure causes a reduction in its mechanical strength, which means that it cannot carry the assumed loads envisaged for full-value wooden materials in a building.

Modern optical systems are based on four-sided scanning of sawn timber surfaces. The use of numerical programs allows an image of defect distribution to be processed, and, as a result, a signal that optimizes the process of sorting sawn timber into construction elements can be generated [19,20,21]. The assessment of defect distribution for all planes of sawn timber allow for the identification of the intensity of undesirable features and linking them with strength. This process consists of processing the image of sawn timber into cross-sectional zones and verifying both the longitudinal and transverse intensity and the distribution of defects according to a predetermined division algorithm. The division formula is influenced by assigning the given parameters to the zones of sawn timber that define their unacceptable ranges. This formula is closely related to the species of wood since the defects for different species are subjected to a different strength classification. Therefore, optimization of the strength sorting process based on optical verification consists of finding a relationship regarding knowledge about the occurrence of defects as a function of the dimensions assigned to zones for different strengths of a considered species of wood. The evaluation process ends by generating a feedback signal, which consequently optimizes the sorting process.

The process of zoning, i.e., automatic detection of significant sections with defects, allows to calculate the area of their occurrence as the ratio of their surface area to the total surface of the timber [22,23,24]. In addition, existing algorithms are able to correctly fit the modulus of elasticity (determined dynamically or sonically) to the intensity of sections on various surfaces of the sawn timber, with an emphasis on knots, slope of fibers, or cracks. Furthermore, it allows to predict the bending strength, which is also related to the share of the selected defects. Adjustment of the determination of the R^2^ model presenting the effect of knots in the discussed algorithms for Douglas fir (*Pseudotsuga enziesii*) is assessed at the level between 0.59 and 0.72, and for Norwegian spruce (*Picea abies*), between 0.42 and 0.50. Studies have shown that determining the area of knots and the dynamic modulus of elasticity can provide an accurate estimate of bending strength values [25,26,27,28].

The currently developed three-dimensional visualization systems for assessing the distribution of defects are based on implemented control systems through a relational data model with four types of parameter classification, i.e., width, length of defects, and their intensity of occurrence. The developed tools are based on a system of comparing the total values of identified defects to their dimensions, shape, or position on both the length and width of a sawn timber piece [19,29]. 

On the basis of nondestructive testing, it is possible to determine the impact of both the quality of raw material and the impact of selected factors on the properties of structural timber intended for specific applications in construction. The correlation between the examined visual features and physical-mechanical properties are the basic indicators determining the suitability of materials for their use in building structures. The bending strength has a strong impact on the design of wooden structures, and it is often determined during nondestructive tests (NDT) with the simultaneous determination of the modulus of elasticity [9,30,31]. Currently, NDT based on ultrasound are one of the most popular methods of various material investigations, and they are used both in scientific research and in various industry branches [32,33,34,35,36,37,38]. The possibility of qualitative (strength) classification of wood based on the propagation speed of ultrasonic and elasticity vibration waves results from the fact that the investigated values depend on the moisture content (MC), the occurrence of defects, species, and the direction of propagation that define its mechanical properties [39,40,41,42,43,44,45,46,47,48]. The use of these methods is associated with the determination of the sonic modulus of elasticity (correlated with the static modulus), which allows for the complete assessment of wood quality [49,50,51]. NDT, however, require considerable knowledge about the properties and structure of wood, the expected distribution of defects and inhomogeneities, as well as the features of the measuring device.

The main purpose of the present work was aimed to confirm the impact of the number and intensity of defects on the planes of sawn timber on the evaluation of its strength parameters. The sonic modulus of elasticity was used as the indicator providing information regarding the technical parameters of timber. The conducted research was aimed at indicating the quality of Polish pine wood and determining its suitability for use in construction. The assessment concerned the verification of the suitability of pine wood from western Poland in industrial practice and the verification of visual methods of classification for coniferous sawn timber.

## 2. Materials and Methods

Pine timber pieces (*Pinus sylvestris* L.) with a dimension of 3500 mm × 175 mm × 22 mm were obtained from large-sized roundwood harvested from mixed forest habitat in the Olesno Forest District (southern Poland, division 14d, 120-year-old stand, fertile soils of podzolic and brown type: clay sands and sandy loams). The habitat type is one of the factors that determine the classification of individual forest areas [52]. The type of forest soil is strictly defined, and it reflects the fertility of the land on which the trees grow. Consequently, it determines the future species composition of the region and both growth and quality development. The appropriate selection of three species, based on the knowledge of their ecological conditions, ensures better land use and natural soil protection [53,54]. 

The present study assessed defects in the anatomical structure and secondary defects (cracks) in the obtained sawn timber. Visual examination of the sawn timber was carried out with the use of a four-sided optical scanner Q-Scan 60^4^, produced by the Polish company Woodinspector form Lublin, Poland. The experimental material was tested by measuring defects, such as knots and disruptions of grains for individual planes, and automatically digitizing and mapping in terms of the frequency of defects present in the examined surface (Figure 1, Figure 2 and Figure 3). Most of the processes in automated timber sorting are related to defect recognition. For crucial defects, proper designations were adopted for knots:knots_0 for sizes up to 10 mm;knots_1 for sizes up to 20 mm;knots_2 for sizes up to 30 mm;knots_3 for sizes over 30 mm;

and for cracks:pek0 for cracks up to 1 mm;pek1 for cracks over 1 mm.

During the analysis, the zones were separated along the length of tested sawn timber and on its cross-section: for wide planes (site_0 and site_1, or both site_0/1) and for narrow planes (site_2 and site_3, or both site_2/3).

The collected information was then compared to the mechanical and physical properties, i.e., modulus of elasticity and apparent density. Strength grading was carried out using an MTG device from the Dutch company Brookhuis Electronics BV (Brookhuis Applied Technologies, Eschede, The Netherlands). The operation of the device is based on measurements of the frequency of wood vibrations caused by dynamic hitting of the end of the tested piece of wood. The device records the time it takes for an acoustic wave to pass through a material of known length, and the propagation speed of the wave can be determined. This speed strongly depends on the structure and damage of the material. In the case of wood, its value is several times higher for the direction of wave propagation along than across the grain. As previously stated in the literature, this method allows to determine the dynamic modulus of elasticity [48]. The MC of wood was determined using a Tanel HIT-3 produced by the Polish company Tanel form Gliwice, Poland, moisture meter with an accuracy of 0.1% just before testing its mechanical properties. During the MC measurements, the nominal density of pine wood and the temperature of the room in which the tests were carried out were assessed and taken into account. The density of each board was determined using the stereometric method (EN 384 [55]). The conversion of *MOE* to a MC of 12% was performed using the Bauschinger formula (Formula (1)):(1)$$MOE_{12}=MOE_{w}\left(1+\alpha_{w}\times \left(w-12\right)\right)$$
where:

*MOE*_12_—modulus of elasticity for MC of 12%, kN/mm^2^;

*MOE_W_*—modulus of elasticity for given MC, kN/mm^2^;

*α_w_*—coefficient of change in modulus of elasticity of wood with a change in its MC by 1%;

*w*—MC of wood during testing, %.

Both the mechanical properties and wood density are summarized in Table 1.

Verification of the relationship between the quality characteristics and the sonic modulus of elasticity consisted of measuring the defects of the sawn timber, including the knotted zone, disruptions of grains and cracks, and their reference in the plane to the confirmed strength evaluation index.

## 3. Results and Discussion

In the conducted research, the relation between the modulus of elasticity and the density of sawn timber with a moisture content of 12% (ρ_12_) was determined. The obtained results show a correlation at a level of R^2^ = 0.804, which can also be described by the following function: *MOE* = −0.0187ρ^2^ + 58.32 ρ − 138,831 (Figure 4). Thus, the analyzed values indicate a high level of fit between the *MOE* and the density of the tested pine timber. Moreover, the Spearman’s rank correlation coefficient was R = 0.896 at a significance level of *p* < 0.001, which shows a high level of correlation between the results of the determined properties.

The effect of the frequency of knot occurrence along the length of the tested pine sawn timber on the *MOE* results was also investigated (Figure 5).

The results confirm the observations previously described by Lin et al. [29], Lukacevic et al. [56], and Wright et al. [57] regarding the effect of knot presence on *MOE* values. In the case of analyzing the effect of the quantity of knots occurring on the wide planes, the Spearman’s rank coefficient was R = −0.67 at a significance level of *p* < 0.001, which can be described by the function y_0/1_ = 3.0198x^2^ − 273.02x + 15,881. At the same time, the Spearman’s rank coefficient was R = −0.30 when analyzing the effect of knot occurrence within the narrow planes on the *MOE* results, and it can be described as follows: y_2/3_ = −2.6031x^2^ + 15.916x +11,953.

In order to specify the effect of knot presence on the length of evaluated sawn timber, measurements of the dimensions of knots were conducted. Based on that, the knots were labeled as 0, 1, 2, and 3. The results are presented in Figure 6.

The outcomes confirm a low correlation between the share of knots and the *MOE* values for those labeled as 0 (up to 10 mm) and 1 (up to 20 mm). At the same time, the negative impact of the increase in the share of the large knots on *MOE* values was noted. The Spearman’s rank correlation coefficient was R = −0.336 for knots labeled as 2 (up to 20 mm) and R = −0.57 for the largest knots labeled as 3 (dimensions over 30 mm).

The occurrence of knots is associated with a change in the anatomical structure of wood. The resulting disruptions of grains may cause a deterioration in the strength parameters, e.g., represented by *MOE* values, as shown in Figure 7. The presented averaged percentage indicators illustrate the major dispersion of the obtained results concerning the frequency of the disruption of grain occurrence along the length of the investigated sawn timber piece. However, at the same time, the distribution of values draws attention to the tendency of the *MOE* results to decrease along with an increase in the share of the disruptions of grains on the surface of the timber. The Spearman’s rank correlation coefficient was R = −0.749 at a significance level of *p* < 0.001. The results confirm previous studies on the direct effect of lumber fiber deviation on strength [17,58,59,60].

Another factor influencing the strength properties of wood is the presence of cracks, which can be called interruptions in the continuity of the material structure. The impact of the number of cracks on individual planes of sawn timber on the *MOE* results is presented in Figure 8. In most cases, cracks in the wood structure are caused by desorption changes. The damage is more intense in the area of the fiber cross-section. Studies have shown that as the number of cracks increase, the values of *MOE* noticeably decrease. This observation confirms the results of previous research conducted by Schajer et al. [26], Olsson et al. [58,61], Burchelt et al. [17], and Yu et al. [62] concerning the effect of the presence of cracks on the timber strength index determined using the modulus of elasticity. The Spearman’s rank correlation coefficient for wide planes labeled as 0 and 1 had slightly negative values of R = −0.342 and R = −0.356, respectively. In the case of analyzing the impact of cracks occurring on both wide planes (0 and 1), there was a strongly negative value of R = −0.741, which indicates a particularly negative effect of their presence on *MOE*.

Despite considerable variability in the distribution of narrow cracks up to 1 mm (labeled as 0) and those characterized by a width of over 1 mm (labeled as 1) in the tested sawn timber, their effect on the *MOE* results was determined (Figure 9). The outcomes did not show an effect of the presence of narrow cracks on the investigated parameter. On the other hand, in the case of wider ones, an adverse effect on the *MOE* results was confirmed. The reason for this is a greater depth of cracks in the structure of timber. Spearman’s rank correlation coefficient for pek0 showed a slightly negative impact of R = −0.342, while the effect of wide cracks (pek1) was strongly negative with R = −0.741.

Verification of the cracks’ surface share in the investigated pine sawn timber allows for the evaluation of secondary damage in the tested piece (Figure 10). The result of the average share of the cracked surface was approx. 0.03%. The distribution of cracks on the planes of timber had a strong negative effect on the outcomes of *MOE*. The Spearman’s rank correlation coefficient was R = −0.824 for the analyzed features.

The investigated effect of the qualitative assessment of sawn timber on the outcomes of *MOE* showed significant roles for both knots with major dimensions and cracks. However, the influence of small knots on the value of *MOE* was not confirmed. Research into the correlation between the area of defects and the *MOE* values of timber obtained from Poland confirms the significant role of this indicator in the assessment of mechanical properties.

Based on the obtained results presenting the technological characteristics of Polish sawn timber, it was possible to assess the impact of defects on the quality of the material. The modulus of elasticity is an indicator of structural timber usability, and it translates into the potential application in this constantly developing branch of industry. As it was shown, its values are related to the share of specific defects in the structure of wood material [25,26,27,28]. In the case of sawn timber of Polish origin, the values of *MOE* are strongly influenced by the dimensions of knots. The effect of the number of knots is much less noticeable. Thus, the research confirmed that it is mainly the share of large knots that is the main factor affecting the results of *MOE*. The share of knots up to 10 mm was not of such importance in the conducted research. Studies have also confirmed a strong negative correlation between *MOE* and the increasing occurrence of the disruptions of grains. The overall research regarding the impact of this defect confirms that there is a strong relationship between the mechanical characteristics of wood and the deviations in grain course resulting from the change in the structure of wood tissue in the area around the knot [63,64]. Not only do the defects resulting from the disturbed anatomy of wood have a significant effect on its strength characteristics, but the secondary defects such as cracks also have a noticeable influence on them. The decrease in wood strength results from the change in the structure of the wood in the area that breaks the continuity of material, and it depends on its extent. It was confirmed that the presence of cracks having larger widths affected the timber properties more [15,17,65,66]. The present work is the basis for further verifications of the structural features of pine wood of Polish origin oriented toward visual strength sorting. The decrease in strength classes could be determined by an increase in the share of defects in terms of their surface area. It is an indicator that also translates into their distribution on a cross-section, which has been previously confirmed by numerous studies using innovative techniques, such as computed tomography [67,68,69,70].

## 4. Conclusions

-The distribution of knots in pine timber grown in western Poland is characterized by a high frequency of their distribution and considerable size. This may indicate a major variability in the quality characteristics of the raw material harvested from the pine stands.-It was examined how the intensity of sound and rotten knots in the tested material translates into the strength indicators. It was found that the share of large knots (over 30 mm) caused the most notable decrease in *MOE* values.-The correlation between the modulus of elasticity and the intensity of knot distribution was positive. There was no evidence of a relationship between half-rotten knots and the tested strength indicator, which, however, taking into account the small share of this defect (approx. 1%), does not confirm a lack of correlation in general.-Despite a significant share of knots with a dimension of up to 20 mm in the pine timber, the effect of their presence on the *MOE* results was not confirmed.-Surface cracks had a slight impact on the *MOE* values of the tested batch of material. On the other hand, the increase in the share of wide cracks occurring on the surface of the sawn timber significantly affected its strength. The research allowed to determine the level of crack area severity for pine materials and the impact of these defects on technical performance (achieving a specific strength).-Along with an increasing level of accuracy in the qualitative characterization of structural sawn timber, it is possible to improve semifinished product sorting efficiency with the use of surface defects analysis. The measurements based on the available image analysis systems allow to locate defects with high accuracy. They are guidelines for the creation of algorithms that allow to determine optimal solutions for the identification of pine wood strength classes.

## Figures and Tables

**Figure 1 materials-15-03974-f001:**
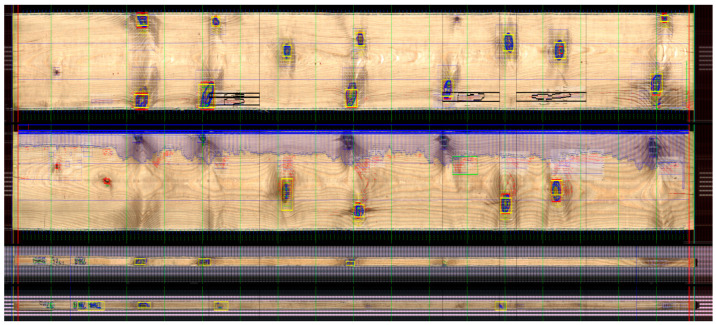
Projection of a digital image of the tested sawn timber.

**Figure 2 materials-15-03974-f002:**
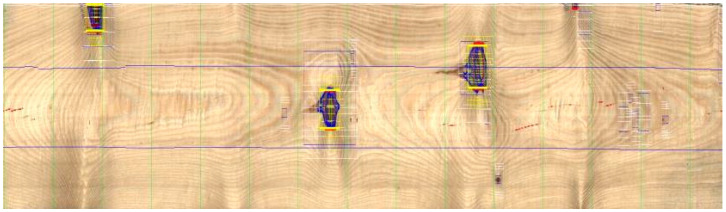
Separation of the zones with knots and disruptions of grains.

**Figure 3 materials-15-03974-f003:**
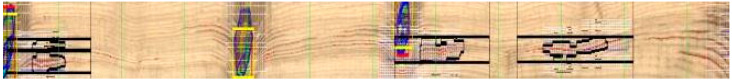
Scheme of cracks zones separation.

**Figure 4 materials-15-03974-f004:**
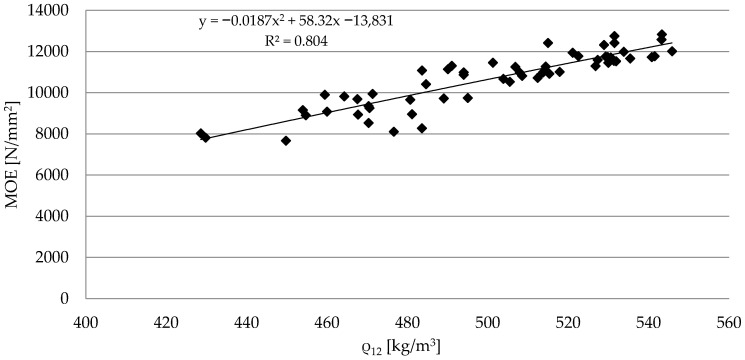
The relation between the modulus of elasticity and density of sawn timber.

**Figure 5 materials-15-03974-f005:**
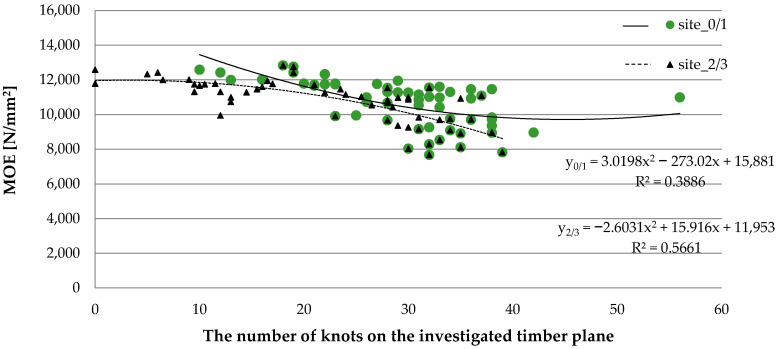
The relation between the modulus of elasticity and knot frequency of sawn timber.

**Figure 6 materials-15-03974-f006:**
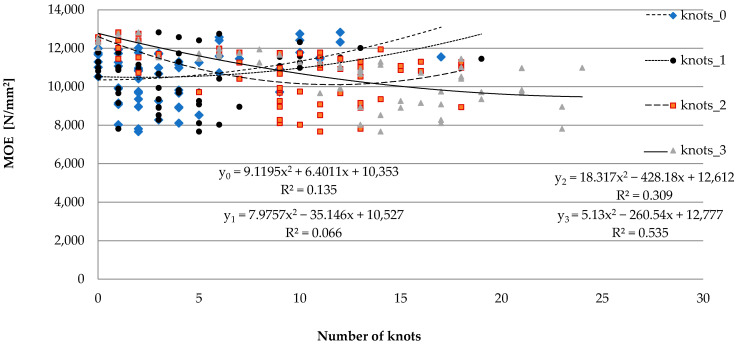
The effect of the share of knots having variable sizes on the *MOE* values.

**Figure 7 materials-15-03974-f007:**
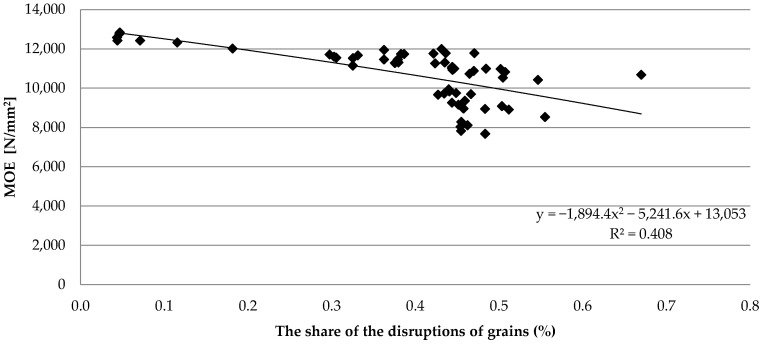
The effect of disruptions of grains on the results of *MOE*.

**Figure 8 materials-15-03974-f008:**
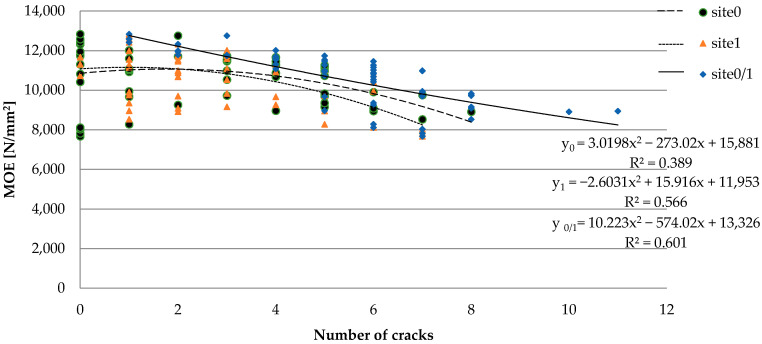
The effect of crack presence on the results of *MOE*.

**Figure 9 materials-15-03974-f009:**
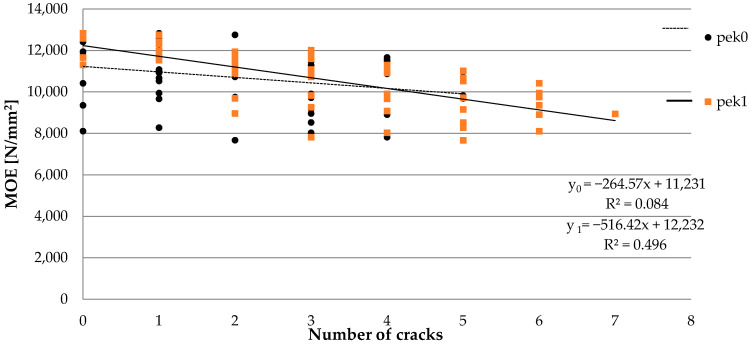
The effect of crack width and number on *MOE* results.

**Figure 10 materials-15-03974-f010:**
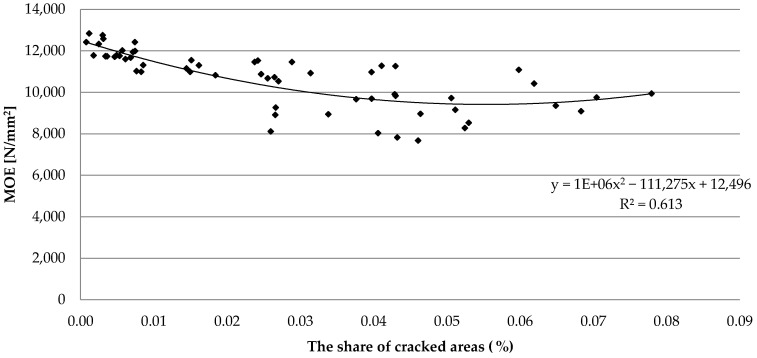
The effect of crack surfaces on *MOE* results.

**Table 1 materials-15-03974-t001:** Properties of research material.

Property	Moisture Content (%)	Density for MC of 12% (kg/m^3^)	*MOE* for MC of 12% (kN/mm^2^)
Average value	13.0	500.7	10,656
Minimum value	11.9	428.6	7674
Maximum value	14.1	545.8	12,835
Standard deviation	0.5	30.3	1344

Note. *MOE*_dyn_, dynamic modulus of elasticity; MC, moisture content.

## Data Availability

The data are included in the paper.

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
