# Peer review of "The Strength of Pine (Pinus sylvestris L.) Sawn Timber in Correlation with Selected Wood Defects"

_materials, 2022, doi:10.3390/ma15113974_

Round 1
Reviewer 1 Report
The problem for the paper is interesting.
I see a lot of problem, the paper can improved
state of art and citated lit. is not sound (poor), missing a lot of basic works about grading (Glos, Steiger, Görlacher and a lot of others). A lot of polish works, not really well known, missing basic publations in this area
Please discribe more basisc the iimportant influecing factors, then the test, please more international. We also have not special "Polish Pine" i, ithink dependeing from growing conditions is similar.
some other comments in the attached file

Author Response
Reviewer 1
Thank You very much for your thorough evaluation of our publication. Your comments and corrections are very valuable. They represent a significant improvement in the quality of the publication. We hope that the present explanations will be satisfactory to You.
With best regards
Authors
The problem for the paper is interesting.
I see a lot of problem, the paper can improved
Reviewer: State of art and citated lit. is not sound (poor), missing a lot of basic works about grading (Glos, Steiger, Görlacher and a lot of others). A lot of polish works, not really well known, missing basic publations in this area
Answer: missing reference to primary works on grading of structural timber evaluation and types of nondestructive testing were completed in the paper
Schickhofer G, Hasewend B, Katzengruber R. Brettsperrholz - Anwendungen und Konstruktionsdetails im mehrgeschossigen Wohn- und Kommunalbau. in Ingenieurholzbau, Karlsruher Tage. Karlsruhe: Bruderverlag. 2002. 285-305
Steiger, R. Mechanische Eigenschaften von Schweizer Fichten-Bauholz bei Biege-, Zug-, Druck- und kombinierter M/N Beanspruchung Sortierung von Rund- und Schnittholz mittels Ultraschall. IBK Bericht 221. Zürich, Juni 1996. https://doi.org/10.3929/ethz-a-001676722
Steiger, R., Gülzow, A., Gsell, D. Non-destructive evaluation of elastic material properties of cross-laminated timber. In: Proceedings of COST E53 conference on “End user’s needs for wood material and products”. 29th–30th October 2008, The Netherlands, pp 171–182
Buchelt, B., Wagenführ, A., Dietzel, A. et al. Quantification of cracks and cross-section weakening in sliced veneers. Eur. J. Wood Prod. 76, 381–384 (2018). https://doi.org/10.1007/s00107-017-1238-z
Martins CEJ, Días AMPG, Marques AFS, Días AMA (2017) Non-destructive methodologies for assessment of the mechanical properties of new utility poles. BioResources 12(2):2269–2283. https://doi.org/10.15376/biores.12.2.2269-2283
Schlotzhauer P, Wilhelms F, Lux C, Bollmus S (2018) Comparison of three systems for automatic grain angle determination on European hardwood for construction use. Eur J Wood Prod 40:118. https://doi.org/10.1007/s00107-018-1286-z
Sandoz JL (1989) Grading of construction timeber by ultrasound. Wood Sci Technol 23(1):95–108. https://doi.org/10.1007/BF00350611
Olsson A, Pot G, Viguier J, Faydi Y, Oscarsson J (2018) Performance of strength grading methods based on fibre orientation and axial resonance frequency applied to Norway spruce (Picea abies L.), Douglas fir (Pseudotsuga menziesii (Mirb.) Franco.) and European oak (Quercus petraea (Matt.) Liebl/Quercus robur L.). Ann For Sci 75(4):33529. https://doi.org/10.1007/s13595-018-0781-z
Reviewer: Please discribe more basisc the iimportant influecing factors, then the test, please more international. We also have not special "Polish Pine" i, ithink dependeing from growing conditions is similar.
Answer: Reviewer is absolutely right. It only takes into account that the pine growing on Polish terrain is characterized by changes depending on the microclimate and type of habitat
Reviewer: Ultrasound is not really often used for grading, mostly vibration (eigenfrequency), for ultrasound we have a quite old PhD work from Rene Steiger, ETH, for eigenfrequency Görlachen (Karlsruhe, both not citated)
Answer: Missing references to basic works on vibration and wave propagation studies are included in the paper
Montero, M.J. Clasificación de madera estructural de gran escuadría de Pinus sylvestris L. mediante métodos no destructivos. (Grading of structural large cross-section timber of Pinus sylvestris L. by nondestructive methods). Doctoral thesis. Universidad Politécnica de Madrid, ETS de Ingenieros Agrónomos 2013. 345 pp. http://oa.upm.es/id/eprint/15201/contents
Tanaka, T., Nagao, H., Nakai, T. Nondestructive evaluation of bending and tensile strength by longitudinal and transverse vibration of lumber. Proceedings of the 8th International Nondestructive Testing of Wood Symposium. Washington State University. Vancouver, Washington 1991, USA, pp. 57–72.
Chauhan, S., Sethy, A. Differences in dynamic modulus of elasticity determined by three vibration methods and their relationship with static modulus of elasticity. Maderas, Cienc. tecnol. 2016, 18(2). doi:10.4067/s0718-221x20160050000
Reviewer: This is vibration test, please explain
Answer: Explanation is provided in the article
Reviewer: The grafic is poor to undertsna, overloaded with informations this give not a influence r2 really low
Answer: Graphic is corrected.
Reviewer: I think we have the same for pine for other countries, this is no special for Polish timber
Answer: Reviewer is absolutely right. It only takes into account that the pine growing on Polish terrain is characterized by changes depending on the microclimate and type of habitat.
Reviewer: Too much Polish papers, this is not really the state of art missing works for Glos (TUMunich);Steiger (ETH, sound propagation); Görlacher (vibration), also works from other poepels like TU Graz etc
Answer: The additional references are now provided.
Görlacher, R. (1984) Ein neues Messverfahren zur Bestimmung des Elastizitätsmoduls von Holz. Holz Roh Werkst. 42:219–222. https://doi.org/10.1007/BF02607231
Görlacher, R. Brettsperrholz: Berechnung von Elementen mit kreuzweise verklebten Brettern bei Beanspruchung in Plattenebene. In: Ingenieurholzbau – Karlsruher Tage. Bruderverlag, Karlsruhe 2002
Wagenführ, A., Buchelt, B. & Pfriem, A. Material behaviour of veneer during multidimensional moulding. Holz Roh Werkst 2006, 64, 83–89. https://doi.org/10.1007/s00107-005-0008-5
Reviewer: Please explain also basic influencing factors
Answer: Explanations are provided.
We thank the Reviewer for important comments that enhance the work.

Reviewer 2 Report
1. There are some grammatical mistakes in the manuscript, please consider correcting them.The language of the whole paper needs to be improved.
2. In the introduction part, there are no references in the whole paragraph(1,5), which needs to be modified and improved.
3. In the whole paper, the author chooses quadratic nonlinear equation to fit the obtained data. Why? The coefficient(a,b,c) of the equation (y=ax2+bx+c) is not explained, why? If the author wish to aquire the higher degree of fit value R2, why not choose the cubic nonlinear equation?
4.The conclusion is part of the text, and too long, which needs to be condensed.
5.The rules and mechanism explanation of the obtained data by the authors is not clear enough, which I feel that the research depth of the manuscript is not high. It is hoped that the author could add some relevant references in the result part in order to compare with the obtained data. If the data have any difference with previous studies, please point out the cause.
6. In Figure 9. the title is"The effect of crack width on MOE results" ,but in the x-coordinate is"Number of cracks". why?
7. All the charts in the manuscript are not standard, please modify.
8. line 212 the description of Figure 5. “The relation between the modulus of elasticity and density of sawn timber ” is wrong.Figure 4 has the same title as Figure 5
Author Response
Reviewer 2
Thank You very much for your thorough evaluation of our publication. Your comments and corrections are very valuable. They represent a significant improvement in the quality of the publication. We hope that the present explanations will be satisfactory to You.
With best regards
Authors
Comments and Suggestions for Authors
Reviewer: There are some grammatical mistakes in the manuscript, please consider correcting them. The language of the whole paper needs to be improved.
Answer: The text has been re-checked for linguistic correctness.
Reviewer: In the introduction part, there are no references in the whole paragraph(1,5), which needs to be modified and improved.
Answer: The following references are provided.
Wang, W., Zhang, Y., Cao, J. et al. Robust optimization for volume variation in timber processing. J. For. Res. 29, 247–252 (2018). https://doi.org/10.1007/s11676-017-0416-5
Schlotzhauer, P.; Kovryga, A.; Emmerich, L.; Bollmus, S.; Van de Kuilen, J.-W.; Militz, H. Analysis of Economic Feasibility of Ash and Maple Lamella Production for Glued Laminated Timber. Forests 2019, 10, 529. https://doi.org/10.3390/f10070529
Olofsson, L., Broman, O., Skog, J., Fredriksson, M. and Sandberg, D. Multivariate product adapted grading of Scots pine sawn timber for an industrial customer, part 1: Method development. Wood Material Science & Engineering 2019, 14(6), 428–436.
Ridley-Ellis, D., Stapel, P. & Baño, V. Strength grading of sawn timber in Europe: an explanation for engineers and researchers. Eur. J. Wood Prod. 2016, 74, 291–306. https://doi.org/10.1007/s00107-016-1034-1
Reviewer: In the whole paper, the author chooses quadratic nonlinear equation to fit the obtained data. Why? The coefficient(a,b,c) of the equation (y=ax2+bx+c) is not explained, why? If the author wish to aquire the higher degree of fit value R2, why not choose the cubic nonlinear equation?
Answer: The quadratic equation through the value of the coefficient a describes the trend of change of the studied parameter. In the case of the apex, it indicates a potential point of change in monotonicity. The authors indicated this function for describing the trend of parameter changes at their high or low r2.
Reviewer: The conclusion is part of the text, and too long, which needs to be condensed.
Answer: Is is shortened
Reviewer: The rules and mechanism explanation of the obtained data by the authors is not clear enough, which I feel that the research depth of the manuscript is not high. It is hoped that the author could add some relevant references in the result part in order to compare with the obtained data. If the data have any difference with previous studies, please point out the cause.
Answer: The description of the results is expanded. The following references are included.
Görlacher, R. Ein neues Messverfahren zur Bestimmung des Elastizitätsmoduls von Holz. Holz Roh Werkst. 1984. 42:219–222. https://doi.org/10.1007/BF02607231
Görlacher, R. Brettsperrholz: Berechnung von Elementen mit kreuzweise verklebten Brettern bei Beanspruchung in Plattenebene. In: Ingenieurholzbau – Karlsruher Tage. Karlsruhe 2002
Schickhofer. G.; Hasewend, B.; Katzengruber, R. Brettsperrholz - Anwendungen und Konstruktionsdetails im mehrgeschossigen Wohn- und Kommunalbau. in Ingenieurholzbau, Karlsruher Tage. Bruderverlag. Karlsruhe 2002. 285-305
Steiger, R. Mechanische Eigenschaften von Schweizer Fichten-Bauholz bei Biege-, Zug-, Druck- und kombinierter M/N Beanspruchung Sortierung von Rund- und Schnittholz mittels Ultraschall. IBK Bericht 221. Zürich 1996
Olsson, A.; Pot, G.; Viguier, J.; Faydi, Y.; Oscarsson, J. Performance of strength grading methods based on fibre orientation and axial resonance frequency applied to Norway spruce (Picea abies L.), Douglas fir (Pseudotsuga menziesii (Mirb.) Franco.) and European oak (Quercus petraea (Matt.) Liebl/Quercus robur L.). Ann. For Scinces 2018, 75(4), 33529. https://doi.org/10.1007/s13595-018-0781-z
Schlotzhauer, P.; Wilhelms, F.; Lux, C.; Bollmus, S. Comparison of three systems for automatic grain angle determination on European hardwood for construction use. Eur. J. Wood Prod. 2018, 40,118. https://doi.org/10.1007/s00107-018-1286-z
Sandoz, J.L. Grading of construction timeber by ultrasound. Wood Sci Technol 1989, 23(1),95–108. https://doi.org/10.1007/BF00350611
Buchelt, B.; Wagenführ, A.; Dietzel, A.; Raßbach H. Quantification of cracks and cross-section weakening in sliced veneers. Eur. J. Wood Prod. 2018, 76, 381–384. https://doi.org/10.1007/s00107-017-1238-z
Steiger, R.; Gülzow, A.; Gsell, D. Non-destructive evaluation of elastic material properties of cross-laminated timber. In: Proceedings of COST E53 conference on End user’s needs for wood material and products. The Netherlands 2008, p 171–182.
Reviewer: In Figure 9. the title is "The effect of crack width on MOE results" ,but in the x-coordinate is"Number of cracks". why?
Answer: It is corrected. Indeed, the figure shows the effect of crack number, but it also concerns the group pek0 for cracks up to 1 mm, pek1 for cracks over 1 mm etc.
Reviewer: All the charts in the manuscript are not standard, please modify.
Answer: Charts are corrected.
Reviewer: line 212 the description of Figure 5. “The relation between the modulus of elasticity and density of sawn timber ” is wrong.Figure 4 has the same title as Figure 5
Answer: It is corrected.
We thank the Reviewer for important comments that enhance the work.

Round 2
Reviewer 1 Report
i think now is ok
Reviewer 2 Report
Most comments and suggestions raised by reviewers have been addressed and the content is also well-revised, so the paper could be accepted.